# Mechanical and Tribological Characteristics of Cladded AISI 1045 Carbon Steel

**DOI:** 10.3390/ma13040859

**Published:** 2020-02-14

**Authors:** Ruslan Karimbaev, Seimi Choi, Young-Sik Pyun, Auezhan Amanov

**Affiliations:** Department of Fusion Science and Technology, Sun Moon University, Asan 31460, Korea; ruslankarimbaev90@gmail.com (R.K.); seimi_ark@naver.com (S.C.); pyoun@sunmoon.ac.kr (Y.-S.P.)

**Keywords:** AISI 1045 carbon steel, roughness, hardness, tribology, additive manufacturing, ultrasonic nanocrystal surface modification

## Abstract

This study introduces a newly developed cladding device, through printing AISI 1045 carbon steel as single and double layers onto American Society for Testing and Materials (ASTM) H13 tool steel plate. In this study, the mechanical and tribological characteristics of single and double layers were experimentally investigated. Both layers were polished first and then subjected to ultrasonic nanocrystal surface modification (UNSM) treatment to improve the mechanical and tribological characteristics. Surface roughness, surface hardness and depth profile measurements, and X-ray diffraction (XRD) analysis of the polished and UNSM-treated layers were carried out. After tribological tests, the wear tracks of both layers were characterized by scanning electron microscopy (SEM) along with energy-dispersive X-ray spectroscopy (EDX). The surface roughness (R_a_ and R_z_) of the single and double UNSM-treated layers was reduced 74.6% and 85.9% compared to those of both the as-received layers, respectively. In addition, the surface hardness of the single and double layers was dramatically increased, by approximately 23.6% and 23.4% after UNSM treatment, respectively. There was no significant reduction in friction coefficient of both the UNSM-treated layers, but the wear resistance of the single and double UNSM-treated layers was enhanced by approximately 9.4% and 19.3% compared to the single and double polished layers, respectively. It can be concluded that UNSM treatment was capable of improving the mechanical and tribological characteristics of both layers. The newly developed cladding device can be used as an alternative additive manufacturing (AM) method, but efforts and upgrades need to progress in order to increase the productivity of the device and also improve the quality of the layers.

## 1. Introduction

The first tool welding system was invented by The Welding Institute (TWI) in the UK in 1991, namely as friction stir welding (FSW) and additive friction stir (AFS) [1]. In principle, these two systems work by producing a relatively solid tool through rotation in order to influence the unstable surface, and then increasing the temperature, leading to the welding of the materials [2]. In recent decades, some studies have concluded that FSW greatly improves mechanical, tensile, and weldability properties, demonstrating significant successful results in various metallic materials [3,4,5,6,7]. The disadvantages of FSW and AFS systems include their very large installations, costliness and impossible working. Importantly, it is well known that materials manufactured by additive manufacturing (AM) have several drawbacks, such as porosity, micro-cracks, residual stress, high surface roughness, etc. Hence, in this study, a newly developed cladding device and post-cladding surface modification treatment were introduced in order to overcome those disadvantages of FSW and AFS systems. The cladding device is a system that was recently developed on the basis of a new working principle and design. The key point of the device is to build a material layer-by-layer by means of rotating filler at high temperature. The cladding device operates by increasing the temperature of the filler to its melting point by means of the rotating movement of three rollers in relative motion. The difference between the newly developed cladding device and the conventional FSW and AFS systems is that it utilizes a filler as a solid tool.

Nowadays, the severe surface plastic deformation (S^2^PD) method plays a vital role in materials science to improve overall properties, especially the yield strength, of metallic materials. For instance, an ultrasonic nanocrystal surface modification (UNSM) treatment is capable of introducing plastic deformation leading to a grain size refinement and inducing a high residual compressive stress at the surface and subsurface layers, while increasing the mechanical properties of metallic materials. In UNSM treatment, there are several treatment parameters such as amplitude, frequency, linear speed, feed rate, static load, and a tip made of hard materials such as silicon nitride (Si_3_N_4_) or tungsten carbide (WC) [8,9]. These parameters need to be optimized to maximize its impact. It has been reported previously that UNSM treatment can improve the surface integrity and mechanical and tribological properties of metallic materials as well as AM metallic materials [10,11,12,13]. Zhao et al. applied UNSM as a pre-treatment to increase the wear and corrosion resistances of steel. It was found that UNSM treatment exhibited a significant improvement compared to polished steel [14]. Cao et al. investigated the influence of UNSM treatment on the fatigue characteristics of AISI 1045 carbon steel, where the presence of surface nanocrystal layers formed after UNSM treatment at the surface layers can retard the initiation of micro-cracks, leading to a delay in fracture [15]. Aerospace applications were one of the first sectors interested in AM carbon steel.

In general, AM methods such as selective laser melting (SLM), powder bed fusion (PBF), direct energy deposition (DED), and selective laser sintering (SLS) are beneficial in terms of more cost-effective production and lower material loss in comparison with traditional bulk materials. However, a newly developed cladding device has several advantages, such as compactness, light weight, affordability, and capability of working flexibly when compared with the aforementioned AM methods. Ma et al. investigated the influence and effectiveness of UNSM treatment on the surface finish of biomedical nitinol alloys manufactured by SLM [16]. It was found that the UNSM treatment significantly enhanced corrosion and wear resistances of biomedical alloys. Moreover, Zhang et al. found that fatigue behavior and mechanical properties of 3D printed titanium alloys manufactured by direct metal laser sintering (DMLS) were improved [17]. Thus, the objective of this study is to demonstrate the possibility of using the newly developed cladding device to build a material layer by layer by means of rotation reaching the melting point of a filler made of AISI 1045 carbon steel, and also to evaluate its mechanical and tribological characteristics. Subsequently, UNSM treatment was applied to AISI 1045 carbon steel layers on ASTM H13 tool steel manufactured by the cladding device to further improve the surface integrity and mechanical and tribological characteristics. The obtained results of this study will contribute to cost, production and material loss of the development of AM systems.

## 2. Materials and Experimental Methods

### 2.1. Test Material

In this present study, a newly developed cladding device was used to print an AISI 1045 carbon steel layer-by-layer onto ASTM H13 tool steel plate. AISI 1045 carbon steel is an excellent steel due to its high specific strength, heat resistance, and good machinability, combined with ductility and viscosity [18,19]. Traditionally, AISI 1045 carbon steel is widely used in the industrial bearings and in other applications such as gears, machine tools, and several mechanical parts [20,21]. Tool steel was selected as a basic material because it is used in industrial knives. The final goal is to clad an ASTM H13 tool steel onto AISI 1045 carbon steel plate in order to reduce the financial expenditures and because of the strategic cost cutting for spin-off DesignMecha Co., Ltd. (Asan, Korea). The chemical composition and essential mechanical properties of AISI 1045 carbon steel and ASTM H13 tool steel are given in Table 1 and Table 2 [22], respectively. The mechanical properties of AISI 1045 and ASTM H13 were provided by a steel supplier OTAI Special Steel Co. Ltd. (Dongguan, China).

### 2.2. Cladding Device

Schematic view of a newly developed cladding device can be seen in Figure 1. In this device, two rectilinear rollers with a diameter of 40 mm support the filler, and one of them has rotation transmitted from the motor, while the third roller, with a diameter of 40 mm, is fixed at specified angle of 15°, which provides a thrust load of 100 N. Thrust load was calculated using a load sensor (DBBP-200, BONGSHIN, Seoul, Korea). The working conditions of the cladding device are listed in Table 3. Moreover, the third roller provides a normal force that controls the filler speed (see Figure 1a). Filler made of AISI 1045 carbon steel was cut from plate by metal lathe and the final dimensions of the filler were: 200 mm in length and 10 mm in diameter. However, it is impossible to use a filler with a diameter of 10 mm due to the large contact area, so the filler diameter was optimized. Figure 2 shows actual pictures of the fillers with different diameters ranging from 2.5 mm to 5.5 mm. As shown in Figure 2a, a filler with a diameter of 5.5 mm failed without welding in plate, while the filler with a diameter of 4.5 mm in Figure 2b demonstrated good suitability for the welding/printing process and was chosen as the optimal filler diameter. Moreover, Figure 2c,d show fillers with diameters of 3.5 mm and 2.5 mm, respectively, which also failed due to their relatively small contact areas. Figure 3 shows SEM images of the top surface of single and double layers. It is clear from Figure 3a that the surface of the single layer is relatively smooth and homogenous, while the surface of the double layer is rough and heterogeneous (see Figure 3b) due to the rough surface of the layer beneath. The surface of the single as-received layer has a better relative surface quality, while the surface of the as-received double layer had a poor-quality surface with uneven curvature. In addition, the cross-sectional SEM images of the single and double layers are shown in Figure 4a,b and Figure 4c,d, respectively. It can be seen that both the single and double layers were deposited well onto the plate, and a delaminated part is magnified in Figure 4b,d. The gap between the single layer and the plate was in the range of 0.8 μm and 7.4 μm, while for the double layer, the gap between the double layer and the plate varied between 0.7 μm and 1.6 μm. Additionally, the thicknesses of the single and double layers were approximately 890 μm and 1800 μm, respectively.

### 2.3. UNSM Treatment

UNSM treatment a new mechanical surface modification cold-forging process. The main principle of UNSM treatment is to introduce a severe plastic deformation that leads to grain size refinement and high compressive residual stress at the surface and subsurface layers, while increasing the mechanical properties of the metallic materials. The UNSM treatment parameters are provided in Table 4. Both the single and double layers were first polished and then subsequently treated by UNSM treatment. Details of UNSM treatment can be found in our previous studies [23,24].

### 2.4. Tribological Test

Tribological characteristics of both layers were evaluated using a tribometer (CSM Instruments, Peseux, Switzerland) against Society of Automotive Engineers (SAE) 52100 bearing steel with a diameter of 7.14 mm under dry conditions. Several essential test conditions are provided in Table 5. The tribological experiments were replicated three times for each specimen due to data scattering.

### 2.5. Characterizations

The surface and cross-sections, wear track and fractography of the layers were observed by scanning electron microscope (SEM) along with energy-dispersive X-ray spectroscope (EDX) (JSM-6610LA, JEOL, Tokyo, Japan). The surface roughness and hardness were measured three and four times using a profilometer (SJ-210, Mitutoyo, Tokyo, Japan) and a micro-Vickers tester at 300 gf and 12 s (MVK E3, Mitutoyo, Tokyo, Japan), respectively. Microstructural changes were analyzed using X-ray diffraction (XRD: D8 ADVANCE, Bruker, Karlsruhe, Germany).

## 3. Results and Discussions

### 3.1. Microstructure

Figure 5 shows SEM images of the polished and UNSM-treated surfaces of the single and double layers. It can be seen from Figure 5a,c that the polished layer demonstrated polishing-induced grooves, while those grooves were eliminated after UNSM treatment, as shown in Figure 5b,d. Also, it can be seen from Figure 5b,d that UNSM treatment produced lines along with bulges due to the movement of scanning forward and backward. Moreover, it is obvious that surface defects such as porosities and cracks—shown by yellow arrows in Figure 5a,c—are present on the surface of both the single and double layers, while the UNSM-treated surfaces of both the single and double layers have less regular defects, meaning that UNSM treatment is capable of removing AM-induced surface defects.

### 3.2. Surface Roughness

Surface roughness parameters play a major role in metallurgical industries and tribology [25]. Figure 6 shows the comparative surface roughness results between six different layers. It is clear from Figure 6 that the surface roughness of the as-received single layer was about R_a_ 3.2 μm and R_z_ 20 μm, whereas the polished and UNSM-treated single layers demonstrated a R_a_ of 0.3 μm and 0.5 μm, and a R_z_ of 2.8 μm and 3.0 μm, respectively. The surface roughness of the as-received double layer was about R_a_ 1.2 μm and R_z_ 8.5 μm, whereas the polished and UNSM-treated double layers demonstrated a R_a_ of 0.4 μm and 0.7 μm, and a R_z_ of 2.0 μm and 4.5 μm, respectively. It is evident that the surface roughness of the as-received single and double layers was significantly reduced after UNSM treatment. The increase in surface roughness of the single and double polished layers after UNSM treatment could have occured due to the high strike numbers, leading to an increase in surface integrity. The reduction in surface roughness of the as-received layer after UNSM treatment may be attributed to the elimination of high peaks and valleys [26]. Moreover, UNSM treatment may reduce the porosity, depending on the size and shape of the pores. This phenomenon also plays a key role in reducing the surface roughness of the as-received layers after UNSM treatment.

### 3.3. Hardness Test

Surface hardness of AM materials plays an important role in selecting materials. The presence of surface porosity and cracks in AM materials are considered to be one of the main issues, because it has been found that reducing the surface porosity leads to an increase in hardness [27]. Figure 7a clearly shows the hardness results of the polished and UNSM-treated single and double layers. The hardness of the polished single layer, which was around 530 HV, increased to approximately 660 HV after UNSM treatment. Furthermore, the polished double layer with a surface hardness of around 540 HV increased to approximately 670 HV after UNSM treatment. Zhang et al. reported that UNSM treatment increased the hardness of the AM material, where UNSM treatment eliminated porosity, leading to an increase in hardness [28]. Furthermore, Kim et al. investigated the effect of UNSM treatment on direct energy deposition (DED) materials [29]. It was found that UNSM treatment increased surface hardness. Moreover, a previous study showed that UNSM treatment increased surface hardness of AISI 1045 carbon steel according to the Hall-Pitch relationship, where grain size refinement is the main mechanism [30]. It is also essential to mention here that UNSM treatment easily led to ferrite refinement as reported by Wu et al. [31]. Figure 7b shows hardness results with respect to depth from the top surface up to 500 μm of the polished and UNSM-treated single layers. It is important to note that hardness of the top surface was lower in comparison with the subsurface. The hardness was significantly increased at a depth of 10 μm, and slightly reduced at a depth of around 350 μm. In Figure 7c,d, the hardness results of polished and UNSM-treated double layers can be seen at the depths of the first and second layers, respectively. The most effective depth in terms of hardness was found to be deeper than 500 μm, as the hardness measurement was made up to 500 μm. As shown in Figure 7c,d, the hardness of the top surface was lower than the subsurface, which may be attributed to the annealing and cooling mechanisms, which are dependent on the temperature of the layers with respect to depth during AM. Moreover, the induced residual stress could be responsible for the lower surface hardness in comparison with the subsurface [32]. The mechanism of the increase in surface hardness with respect to depth is a result of grain size refinement and work hardening induced by plastic strain [33]. Specifically, in AM materials, the elimination of surface defects such as porosities, cracks, etc., may also be responsible for the increase in surface hardness with respect to depth [34].

### 3.4. XRD Results

Figure 8a shows the comparison of the XRD patterns of the polished and UNSM-treated single and double layers. The intensity of the XRD peaks of the single UNSM-treated layer was decreased, and shifted towards a lower diffraction angle in comparison to the single polished layer, as shown in Figure 8b. It is also evident that the intensity of the peak the UNSM-treated double layer was increased and shifted towards a lower diffraction angle. Figure 8c shows that the intensity of the peak of the UNSM-treated single layer was decreased and shifted towards a higher diffraction angle, while the UNSM-treated double layer was increased and shifted towards a lower diffraction angle. At the same time, it is clear from Figure 8d that the intensity peaks of the UNSM-treated single and double layers increased with respect to the polished single and double layers and also shifted towards a lower diffraction angle. It is widely known that mechanical surface modification processes can lead to peak shifts to a lower diffraction angle by increasing uniform compressive strain, while it is also important to mention here that higher diffraction angles can be caused through grain reorientation and nonuniform compressive strain [33,35]. Moreover, 2θ position and full width at half maximum (FWHM) value information obtained from XRD patterns of the polished and UNSM-treated single and double layers are provided in Table 6. It was found that FWHM values of both the single and double layers were widened after UNSM treatment, which is an indication of the presence of a nanocrystal surface layer [36]. In general, UNSM treatment tends to produce a nanocrystal surface layer with nano-grain size on the top surface with a thickness of about 100 µm for carbon steels [37].

### 3.5. Variation in Tribological Characteristics

A comparison of the tribological test results of the polished and UNSM-treated single and double layers are shown in Figure 9. It can be seen from Figure 9a that the friction coefficient of the UNSM-treated single layers was lower in comparison with the polished single layer at the running-in period, but continuing the test the friction coefficient of both the layers reached to a steady-state period, where the UNSM-treated single layer demonstrated a slightly higher friction coefficient in comparison with the polished single layer. Figure 9b shows the friction coefficient of the polished and UNSM-treated double layers. It was found that UNSM-treated double layer exhibited a lower friction coefficient in comparison with the polished double layer throughout the sliding cycles. The friction coefficient of the UNSM-treated double layer was very low at the onset of the tribological test with a friction coefficient of 0.0024, and then it increased rapidly to a friction coefficient of approximately 0.24. With continuing the tribological test the friction coefficient of started increasing up to a friction coefficient of 0.27 throughout the sliding cycles. The average friction coefficients of the polished and UNSM-treated single and double layers were found to be 0.246 and 0.228, and 0.267 and 0.214, respectively. The rapid increase in friction coefficient of the polished and UNSM-treated single and double layers may be definitively attributed to the initial surface roughness (see Figure 6) at the initial asperity contacts in relative motion, respectively, whereas rapid running-in and steady-state periods in the friction coefficient are due to the minor wear of asperities after several cycles. Moreover, UNSM treatment can generate high plastic deformation and a nanostructured surface layer. The friction coefficient of the polished and UNSM-treated double layers exhibited a lower value in comparison with the polished single layer throughout the sliding cycles due to the higher initial surface roughness in comparison with that of the single layer, while the surface hardness of both layers was similar. The influence of surface roughness can be explained by the fact that the contact area between the polished surface and the ball increased gradually with increasing reciprocating time. It is clear that increasing contact area between the polished specimen and the ball unfavorably affected the behavior of the friction coefficient. It should be mentioned here that the UNSM treatment reduced the friction coefficient as the shear stress at the contact interface decreased, and also that the beneficial friction coefficient can be attributed to the high plastic deformation and the increase in surface hardness due to grain refinement and work-hardening effects.

The wear rates of the polished and UNSM-treated single and double layers are shown in Figure 10. The wear rate of the polished single and double layers was reduced by about 11% and 19%, respectively, by the application of UNSM treatment. The wear rate of the UNSM-treated single and double layers was found to be higher than those of the polished single and double layers. The reduction in friction coefficient of both the UNSM-treated single and double layers can mainly be attributed to the increase in surface roughness, where the increased hardness is responsible for the increase in wear resistance. As mentioned previously, the friction coefficient and wear rate may be affected by the real contact area due to high surface roughness [26,38]. The friction coefficient reduction and wear resistance enhancement of AM materials by UNSM treatment can be found in previous studies [13,16,29].

### 3.6. Wear Track Analysis

Comparison based on SEM mapping together with oxide distribution over the worn out surface within the wear track generated after tribological testing on the surface of the polished and UNSM-treated single and double layers is shown in Figure 11. Obviously, with respect to the wear track width and depth, both the polished single and double layers were wider and deeper than those of the UNSM-treated single and double layers, as shown in Figure 11a–d. The narrower and shallower wear track of the UNSM-treated single and double layers is mainly attributed to the increase in surface hardness, while an increased surface roughness as a result of UNSM treatment reduces the true contact area of the ball and layer in relative motion. The wear mechanisms of carbon steel sliding against bearing steel under dry sliding conditions were abrasive and oxidative, where the level of oxidation was different for each layer. Figure 11(a1,b1,c1,d1) show the distribution of oxide over the wear track of the polished and UNSM-treated single and double layers. It was found that an oxide film was formed within the wear track of the polished and UNSM-treated single and double layers with oxidation amounts of 10.64, 12.13, 9.68 and 3.97 wt.%, respectively. In general, the formation of oxide film as a tribo-film is known to improve the tribological properties of mating materials by preventing direct contact between the metals [39]. It is noted here that the amount of oxidation within the wear track of the polished single layer was lower in comparison with the UNSM-treated single layer, while the amount of oxidation within the wear track of the polished double layer was higher in comparison with the UNSM-treated double layer. This is related to the surface roughness of both the UNSM-treated layers, where the surface irregularities were large and the curvature was so severe that the contact between the ball and layer was insufficient, so that the formation of wear track was irregular, as shown in Figure 11b,d. The oxide distribution in the wear track was not uniform due to the initial rough surface of both of the UNSM-treated layers. The smaller increase in oxidation amount within the wear track of the UNSM-treated single layer may play an important role in controlling the tribological properties, in particular in reducing the friction reduction and the improvement of abrasion resistance by oxide film. Oxides on the surface can act as a tribo-layer at the contact interface. In addition, SEM-EDX of the counter surface ball was characterized. It was found that no significant differences in wear scar and oxidation were found for the polished and UNSM-treated single layers, while the polished double layer exhibited a bit higher level of oxidation in comparison with the UNSM-treated double layer.

## 4. Conclusions

This study presented the mechanical and tribological characteristics of AISI 1045 carbon steel placed onto ASTM H13 tool steel plate as single and double layers by a newly developed cladding device. The findings of this investigation led to the following conclusions:

Surface roughness of the as-received single layer was about R_a_ 3.2 μm, whereas the polished and UNSM-treated single layers demonstrated a R_a_ of 0.3 μm and 0.5 μm, respectively. The surface roughness of the as-received double layer was about R_a_ 1.2 μm, whereas the polished and UNSM-treated double layers demonstrated a R_a_ of 0.4 μm and 0.7 μm, respectively.

The hardness of the polished single layer, which was around 530 HV, increased to 660 HV after UNSM treatment. The polished double layer with a surface hardness of around 540 HV increased to around 670 HV after UNSM treatment.

The intensity of the XRD peaks of the single UNSM-treated layer was reduced and shifted towards a lower diffraction angle in comparison to the single polished layer. It is also evident that the intensity of the peak of the double UNSM-treated layer was increased and shifted towards a lower diffraction angle.

The friction coefficient of the UNSM-treated single layer was lower in comparison to the polished single layer during the running-in period. The UNSM-treated double layers exhibited a lower friction coefficient in comparison with the polished double layer throughout the sliding cycles.

The wear rate of the polished both single and double layers was reduced by the application of UNSM treatment by about 11% and 19%, respectively. The wear rate of both the UNSM-treated single and double layers was found to be higher than those of the polished single and double layers.

A newly developed cladding device can be used as an AM, but a lot of efforts and upgrades need to be undertaken in order to improve the quality of the layers. Furthermore, UNSM treatment was demonstrated to be applicable as a post-AM surface modification technology for improving the surface quality of AM materials manufactured by cladding device. It is necessary to upgrade the design of the device in order to increase its productivity and to reduce financial expenditures and because of strategic cost cutting for DesignMecha Co., Ltd.

## 5. Patents

A patent related to the filler-based AM system presented in this manuscript has already been filed (No.: 10-2019-0121686).

## Figures and Tables

**Figure 1 materials-13-00859-f001:**
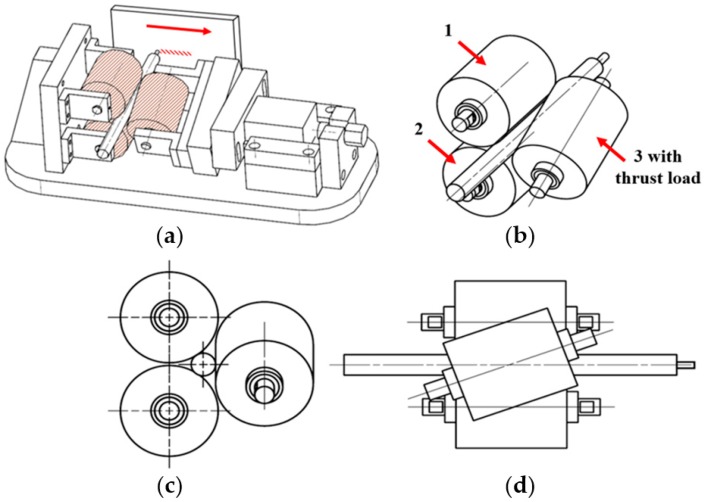
A newly developed cladding device (**a**) and various views of the rotation system (**b**–**d**).

**Figure 2 materials-13-00859-f002:**
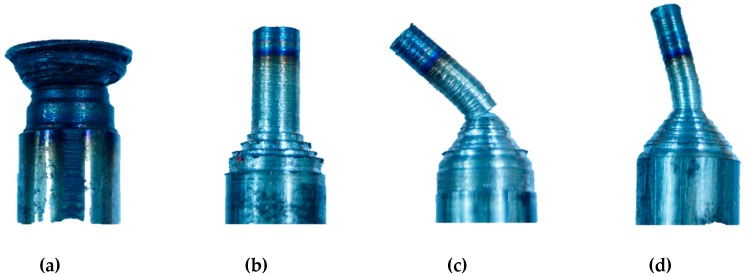
Pictures of fillers with four different diameters: 5.5 mm (**a**), 4.5 mm (**b**), 3.5 mm (**c**) and 2.5 mm (**d**).

**Figure 3 materials-13-00859-f003:**
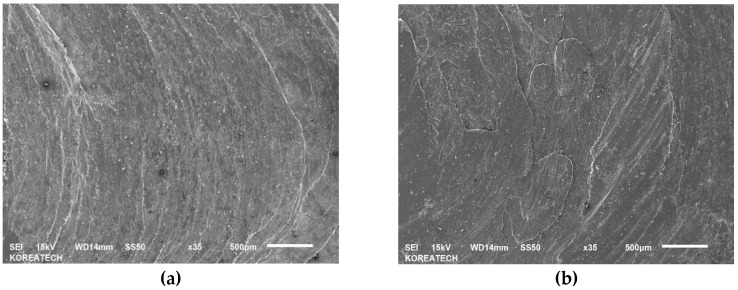
SEM images of the top surface of (**a**) single and (**b**) double layers.

**Figure 4 materials-13-00859-f004:**
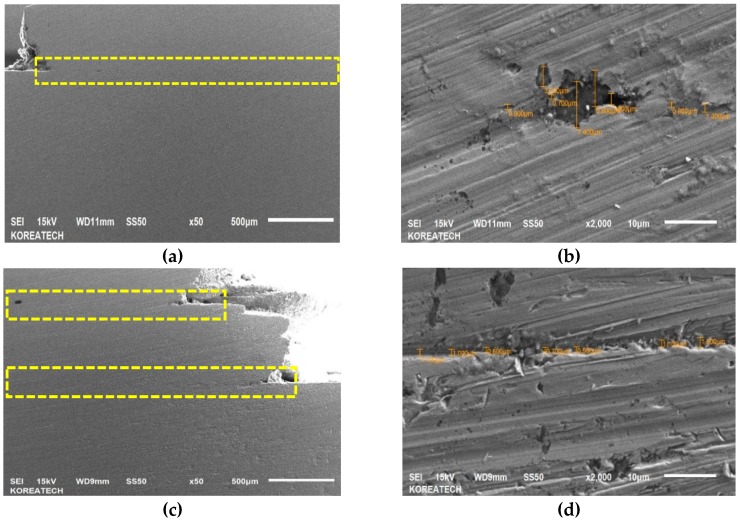
Cross-sectional SEM images of the single (**a**,**b**) and double (**c**,**d**) layers.

**Figure 5 materials-13-00859-f005:**
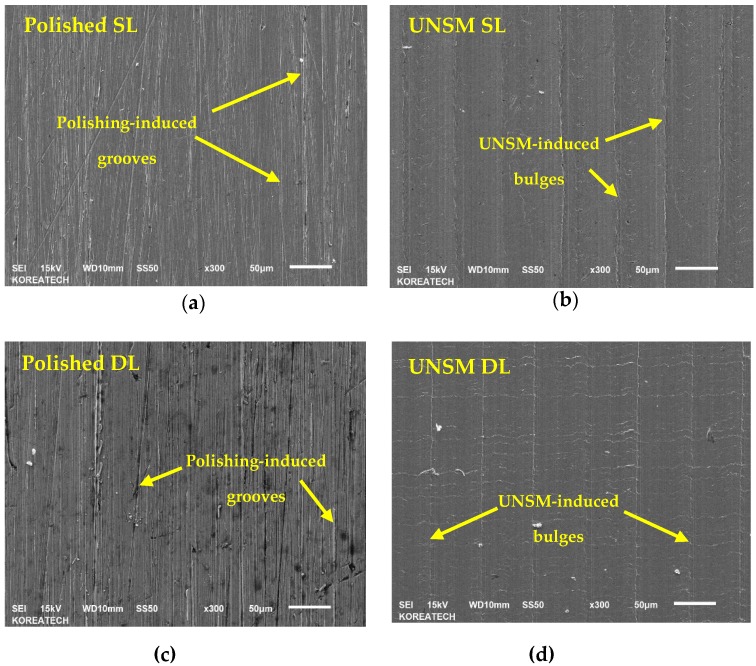
SEM images of the polished and UNSM-treated surfaces of the single (**a**,**b**) and double (**c**,**d**) layers.

**Figure 6 materials-13-00859-f006:**
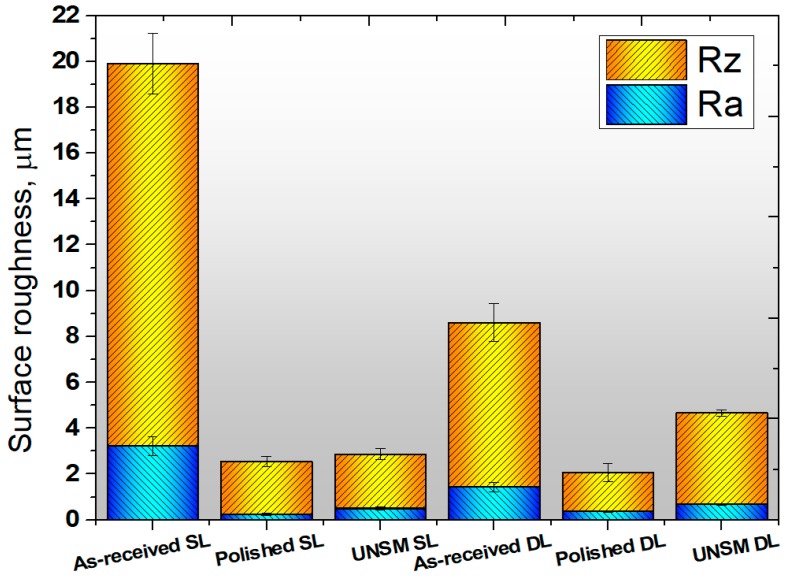
Comparison of surface roughness results of the as-received, polished and UNSM-treated surfaces of the single and double layers.

**Figure 7 materials-13-00859-f007:**
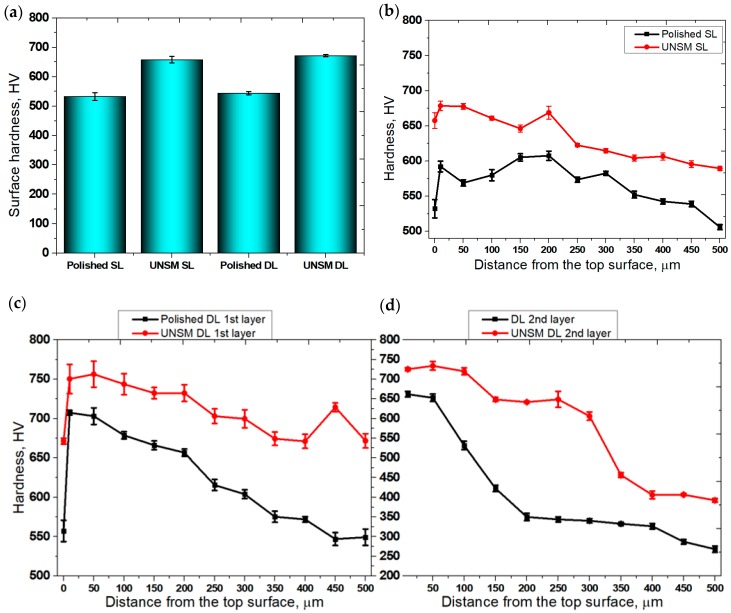
Comparison of surface hardness (**a**) and hardness with respect to depth of the polished and UNSM-treated single (**b**) and double (**c**,**d**) layers.

**Figure 8 materials-13-00859-f008:**
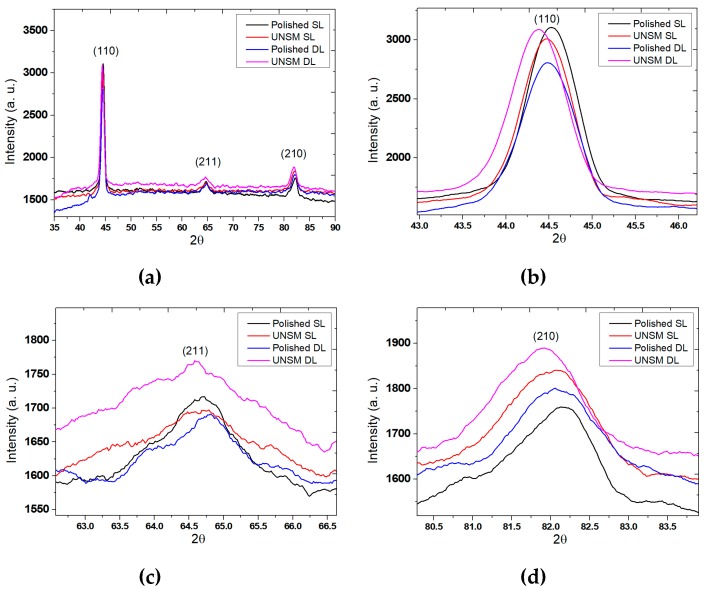
XRD patterns of the polished and UNSM-treated single and double layers (**a**). Primary peak (**b**), secondary peak I (**c**) and secondary peak II (**d**).

**Figure 9 materials-13-00859-f009:**
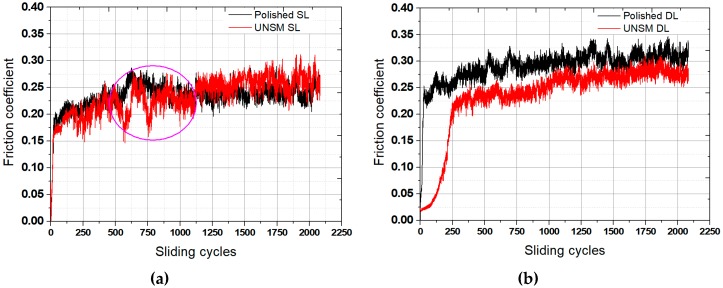
Friction coefficient results of the polished and UNSM-treated single (**a**) and double (**b**) layers.

**Figure 10 materials-13-00859-f010:**
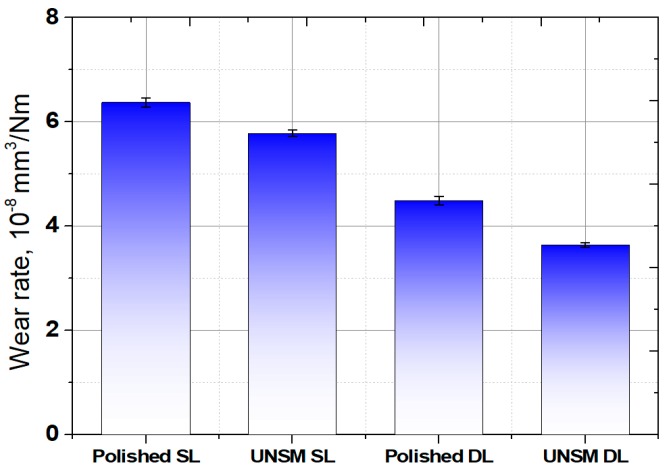
Wear rate results of the polished and UNSM-treated single and double layers.

**Figure 11 materials-13-00859-f011:**
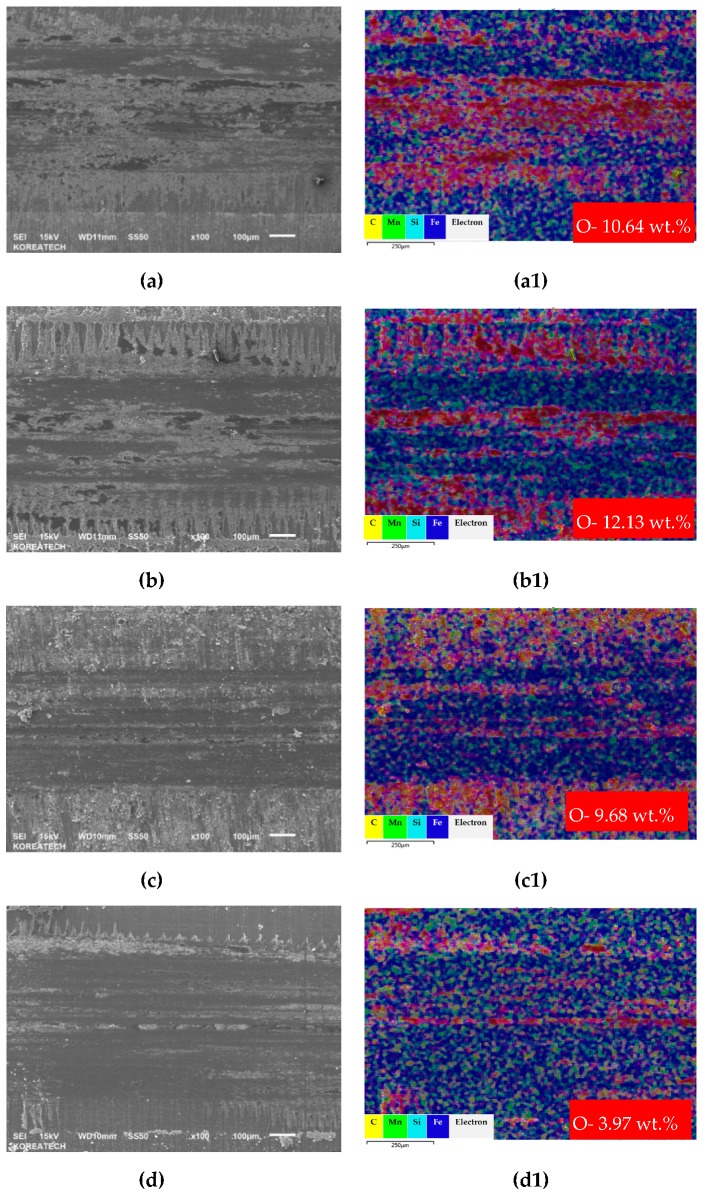
SEM images and mappings of the wear track for the polished (**a**,**a1**,**c**,**c1**) and UNSM-treated (**b**,**b1**,**d**,**d1**) surfaces for both single and double layers.

**Table 1 materials-13-00859-t001:** Chemical composition of AISI 1045 carbon and ASTM H13 tool steels in (wt.%).

Materials	C	Mn	P	S	Si	Cr	V	Mo
AISI 1045	0.50	0.80	0.035	0.035	0.27	0.25	-	-
ASTM H13	0.45	0.6	0.03	0.03	1.25	5.5	1.2	1.75

**Table 2 materials-13-00859-t002:** Mechanical properties of AISI 1045 carbon and H13 tool steels.

Materials	Tensile Strength, MPa	Yield Strength, MPa	Elastic Modules, GPa	Hardness, HV
AISI 1045	569	343	80	260
ASTM H13	1590	1380	215	205

**Table 3 materials-13-00859-t003:** The main parameters of the cladding device.

Roller Diameter, mm	Speed of Roller, rpm	Speed of Filler, rpm
40	5190	20760

**Table 4 materials-13-00859-t004:** UNSM treatment parameters.

Frequency, kHz	Amplitude, μm	Speed, mm/min	Static Load, N	Interval, μm	Ball Diam., mm	Ball Material
20	30	2000	50	0.03	2.38	WC

**Table 5 materials-13-00859-t005:** Tribological test conditions.

Load, N	Frequency, Hz	Sliding Distance, m	Time, min	Temperature, °C
10	1.33	25	26	RT

**Table 6 materials-13-00859-t006:** 2θ position and full width at half maximum (FWHM) value information obtained from XRD patterns of the polished and UNSM-treated single and double layers.

**Single Layer (SL)**			
**Diffraction Peaks**	**Specimens**	**2θ Degree**	**FWHM**
(110)	Polished	44.53	0.5562
	UNSM-treated	44.47	0.6720
(211)	Polished	64.61	1.2822
	UNSM-treated	64.56	2.0780
(210)	Polished	82.03	1.0892
	UNSM-treated	81.95	1.3206
**Double Layer (DL)**			
(110)	Polished	44.48	0.7288
	UNSM-treated	44.37	0.7293
(211)	Polished	64.69	1.3933
	UNSM-treated	64.43	2.8394
(210)	Polished	82.06	1.1465
	UNSM-treated	81.83	1.2285

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
