# Peer review of "Mechanical and Tribological Characteristics of Cladded AISI 1045 Carbon Steel"

_materials, 2020, doi:10.3390/ma13040859_

Round 1
Reviewer 1 Report
The article is valuable because the test results can provide information about the properties of the tested coatings that were created by a newly developed cladding device.
The weakness of the article is the lack of an attempt to explain the observed differences in the properties of the tested materials in the conclusions. Currently, the conclusions are the only summary of the test results. If possible, I would suggest explaining the differences in properties of the tested materials, e.g. why the friction coefficient of the polished and UNSM-treated double layers exhibited lower value in comparison with the polished double layer throughout the sliding cycles?
Another weakness may be that the only statistical information about test results is in the form of error bars on the charts. It is not given what they represent (standard deviation, confidence interval or another parameter?). I suggest supplementing that information. Additionally, I suggest presenting statistical data in the text of the article, such as the number of repetitions of experiments for a given measurement point, standard deviation or confidence intervals, if possible.
The third suggestion is to correct the way of roughness parameters presentation. The values of the parameters describing the surface roughness are usually given in the order: parameter symbol followed by its value and not vice versa (e.g. Ra3.2 µm and not 3.0 Ra) and the unit (µm) is given in the end.
There is a mistake in the sentence (line 302-304) “The friction coefficient of the polished and UNSM-treated double layers exhibited a lower friction coefficient in comparison with the polished double layer throughout the sliding cycles.”
Author Response
Reviewer #1: Comments and Suggestions for Authors
The article is valuable because the test results can provide information about the properties of the tested coatings that were created by a newly developed cladding device.
RESPONSE: Thank You for Your comment.
The weakness of the article is the lack of an attempt to explain the observed differences in the properties of the tested materials in the conclusions. Currently, the conclusions are the only summary of the test results. If possible, I would suggest explaining the differences in properties of the tested materials, e.g. why the friction coefficient of the polished and UNSM-treated double layers exhibited lower value in comparison with the polished double layer throughout the sliding cycles?
RESPONSE: We agree with this comment and explained the mechanism in the revised paper.
Another weakness may be that the only statistical information about test results is in the form of error bars on the charts. It is not given what they represent (standard deviation, confidence interval or another parameter?). I suggest supplementing that information. Additionally, I suggest presenting statistical data in the text of the article, such as the number of repetitions of experiments for a given measurement point, standard deviation or confidence intervals, if possible.
RESPONSE: In this study, all experimental tests and measurements were replicated at least three times. It was mentioned, in Parts 2.4 and 2.5 of the revised manuscript.
The third suggestion is to correct the way of roughness parameters presentation. The values of the parameters describing the surface roughness are usually given in the order: parameter symbol followed by its value and not vice versa (e.g. Ra 3.2 µm and not 3.0 Ra) and the unit (µm) is given in the end.
RESPONSE: The parameter symbol position has been corrected accordingly and units have been added throughout the revised manuscript.
There is a mistake in the sentence (line 302-304) “The friction coefficient of the polished and UNSM-treated double layers exhibited a lower friction coefficient in comparison with the polished double layer throughout the sliding cycles.”
RESPONSE: The sentence has been revised in the revised manuscript. Thank you for Your kind attention.

Reviewer 2 Report
Dear Authors,
I have read Your work with great attention. Your article concerns the application of a new device (developed and patented by the authors of the manuscript) to manufacture a layer of carbon steel on tool steel. The work is a closed research project completed with correctly formulated conclusions. I am convinced that the research material described in the submission will be an important source of information for industrial engineers and scientists. While reading the article, I had a few comments that, hopefully, will help shape the final version of your article.
In my opinion, the article has a generally positive impression, but its reading is hampered by a large number of grammatical errors. I recommend performing a language correction of the entire article. There are many phrases at work that raise grammatical doubts, e.g. line 103, line 261, line 279.
I strongly suggest to change the title of submission. In its present form, the title suggests that mechanical and tribological characteristics of AISI 1045 steel were tested by your device.
Abstract should be supplemented with information on the results obtained (quantitative).
Line 80: why tool steel was used as the basic material. What is its chemical composition? Do differences in the chemical composition of both materials affect the tested properties of the layers?
Table 1: give the elements in the standard order.
Fig. 1: add a), b), c) and d) for better clarity of figure. The more that Figure 1a) is given in line 96.
Table 3: remove plate and filler material columns. These are not parameters.
Line 138: add space before s.
Line 144 and further: add space in: “Figure 5(a and c)”; Figure 5(b and d)”…”Figure 11(a,b,c and d)”.
Figure 7d: the title of the X axis is different than in b and c.
Line 295: correct “ofaround”
The conclusions are well worded. References are well chosen.
Author Response
Reviewer #2: Comments and Suggestions for Authors
Dear Authors,
I have read Your work with great attention. Your article concerns the application of a new device (developed and patented by the authors of the manuscript) to manufacture a layer of carbon steel on tool steel. The work is a closed research project completed with correctly formulated conclusions. I am convinced that the research material described in the submission will be an important source of information for industrial engineers and scientists. While reading the article, I had a few comments that, hopefully, will help shape the final version of your article.
RESPONSE: Thank you for Your attention and valuable comments to improve the quality of the manuscript.
In my opinion, the article has a generally positive impression, but its reading is hampered by a large number of grammatical errors. I recommend performing a language correction of the entire article. There are many phrases at work that raise grammatical doubts, e.g. line 103, line 261, line 279.
RESPONSE: The English has been proofread by a native English speaker.
I strongly suggest to change the title of submission. In its present form, the title suggests that mechanical and tribological characteristics of AISI 1045 steel were tested by your device.
RESPONSE: The title of the manuscript has been changed in the revised manuscript.
Abstract should be supplemented with information on the results obtained (quantitative).
RESPONSE: The abstract has been supplemented with quantitative results in the revised manuscript.
Line 80: why tool steel was used as the basic material. What is its chemical composition? Do differences in the chemical composition of both materials affect the tested properties of the layers?
RESPONSE: In this study, our first step was used to carbon steel as filler onto tool steel. Tool steel was selected as a basic material because it is used in industrial knives. The final goal is to clad an ASTM H13 tool steel onto AISI 1045 carbon steel plate in order to reduce financial expenditures and because of strategic cost cutting for DesignMecha Co., Ltd. Moreover, both materials chemical composition totally different. Chemical composition of tool steel was added in the revised manuscript.
Table 1: give the elements in the standard order.
RESPONSE: In Table 1, the position of chemical elements has been ordered accordingly based on the standard designation ASTM A 29/A 29M-03 in the revised manuscript. Thank you for Your attention.
Fig. 1: add a), b), c) and d) for better clarity of figure. The more that Figure 1a) is given in line 96.
RESPONSE: In Figure 1, a), b), c) and d) have been added for better clarification in the revised manuscript.
Table 3: remove plate and filler material columns. These are not parameters.
RESPONSE: Table 3 has been revised in the revised manuscript.
Line 138: add space before s.
RESPONSE: The space has been added before "s" in the revised manuscript.
Line 144 and further: add space in: “Figure 5(a and c)”; Figure 5(b and d)”…”Figure 11(a,b,c and d)”.
RESPONSE: The spaces have been added throughout the revised manuscript.
Figure 7d: the title of the X axis is different than in b and c.
RESPONSE: You are absolutely right, we are sorry for this mistake. The title in Figure 7d of the X axis has been changed in the revised manuscript.
Line 295: correct “ofaround”
RESPONSE: The sentence has been corrected in the revised manuscript.
The conclusions are well worded. References are well chosen.
RESPONSE: Thank you for Your comment.

Reviewer 3 Report
Comment 1
Line 208 page8/15- ”… Figure 8(d) that intensity peaks of the single and double UNSM-treated
layers were increased” since peak do not show relative intensities this sentence shall be removed or modified.
Comment 2
Fig 11 – element legends are blur, please increase the resolution or write separately in legend. -
Author Response
Reviewer #3: Comments and Suggestions for Authors
Comment 1
Line 208 page8/15- ”… Figure 8(d) that intensity peaks of the single and double UNSM-treated
layers were increased” since peak do not show relative intensities this sentence shall be removed or modified.
RESPONSE: In Figure below, it can be seen that the comparison of XRD patterns between polished and UNSM treated both the single and double layers. It is clear from Figure that the UNSM-treated layer increased intensity and shifted lower compared to the polished one. The sentence has been modified in the revised manuscript.
Comment 2
Fig 11 – element legends are blur, please increase the resolution or write separately in legend. -
RESPONSE: In Figures 11 (a1), 11 (b2), 11 (c3) and 11 (d4) legends resolution have been increased in the revised manuscript.

Reviewer 4 Report
The authors presented a new cladding device with tribological and microhardness tests. Although the essence of this investigation shows a promising manufacturing method, it needs, in my view, a more extended work. I would like to give some suggestions to the authors:
An English spell check is required. There are repeated words like "...the wear track of the of the polished…" and some other style problems. The essence of this investigation is the new cladding device, but there is a lack of information: what was the thickness of the single and double layers? What was the load obtained during the process? This manufacturing process is a novel method, so it would be very interesting to measure this load. The mechanical properties of the clad was quantified with a microhardness test, but I believe that this test only measures the hardness of the clad, not if this clad has been correctly joined to the base material. Figure 4 shows welding defects which could be the origin of cracks and delaminations. It should be tested in fatigue to demonstrate the goodness or deficiency of this novel method. In chapter 2.3, Table 4 is mentioned, but it should mention the table 3. Reference 20 is mentioned as an example of the AISI 1045 application for gears, tools , etc, but this reference does not mention it. The mechanical properties of table 2, how has it been obtained? Extracted from some reference? Please, include it. Tested by the authors? Please, mention it. Reference 18 does not exist.In conclusion, I recommend to reject this paper, because it needs more experimental tests and editing of English language.
Author Response
Reviewer #4: Comments and Suggestions for Authors
The authors presented a new cladding device with tribological and micro-hardness tests. Although the essence of this investigation shows a promising manufacturing method, it needs, in my view, a more extended work. I would like to give some suggestions to the authors:
An English spell check is required. There are repeated words like "...the wear track of the of the polished…" and some other style problems.
RESPONSE: The English has been checked throughout the revised manuscript by a native English speaker.
The essence of this investigation is the new cladding device, but there is a lack of information: what was the thickness of the single and double layers?
RESPONSE: In this study, the thickness of the single and double layers without polishing was from 890 µm to 900 µm, 1790 µm to 1800 µm, respectively. These information have been included in the revised manuscript.
What was the load obtained during the process? This manufacturing process is a novel method, so it would be very interesting to measure this load.
RESPONSE: In this study, the load provided by thrust roller was 100 N that calculation with sensor system as (DBBP-200, BONGSHIN, Korea). Also, this information has been added to the revised manuscript.
The mechanical properties of the clad was quantified with a microhardness test, but I believe that this test only measures the hardness of the clad, not if this clad has been correctly joined to the base material.
RESPONSE: In this study, micro-hardness has been measured at the top surface and with respect to depth. Bonding of clad to the base material will be evaluated by micro-scratch testing as a future study, which mainly will focus on bonding quality and fatigue behavior of cladded layers.
Figure 4 shows welding defects which could be the origin of cracks and delaminations. It should be tested in fatigue to demonstrate the goodness or deficiency of this novel method.
RESPONSE: Yes, we agree with Your comments about fatigue test. In this study, we focus on mechanical and tribological characteristics of AISI 1045 carbon steel. As a next step, we will clad an hourglass fatigue specimens for rotary bending fatigue (RBF) tester machine to investigate the fatigue behavior of aluminum, carbon steel and tool steel materials, which have already been cladded by our new device.
In chapter 2.3, Table 4 is mentioned, but it should mention the table 3.
RESPONSE: You are absolutely right, we are sorry for this mistake. In chapter 2.3, the correction has been made in the revised manuscript.
Reference 20 is mentioned as an example of the AISI 1045 application for gears, tools, etc., but this reference does not mention it.
RESPONSE: As it can be seen from Reference 20 that carbon steel S45C (JIS standard) (also widely known as AISI 1045 (ASTM standard)). In this regard, we mentioned in our manuscript as "Traditionally, AISI 1045 carbon steel is widely used in bearing industries and other applications such as gears, machine tools and several mechanical parts [20]." In addition, we have added additional references in the revised manuscript.
The mechanical properties of table 2, how has it been obtained? Extracted from some reference? Please, include it. Tested by the authors? Please, mention it.
RESPONSE: The mechanical properties in Table 2 have not been calculated by the authors. The mechanical properties of AISI 1045 and ASTM H13 have been provided by a supplier OTAI Special Steel Co. Ltd. (China). References for Table 2 have been included in the revised manuscript.
Reference 18 does not exist.
RESPONSE: We present a screenshot of Reference 18 obtained from ScienceDirect.
In conclusion, I recommend to reject this paper, because it needs more experimental tests and editing of English language.
RESPONSE: The English has been improved throughout the manuscript. The authors hope that the manuscript is now acceptable for publication after substantial improvement.

Round 2
Reviewer 2 Report
Dear Authors,
thank you very much for taking my comments into account. I think the submission can be published in its current form.
During the proofreading, pay attention to the font quality on Figs. 6-10. In addition, in Figure 7, "(c)" appears twice and the title of the Y axis in Figure 7 d) is not visible.
Best regards
Reviewer 4 Report
After these improvements, this article could be published in the present form.